# Navigating through COVID-19 Pandemic Period in Implementing Quality Teaching and Learning for Higher Education Programmes: A Document Analysis Study

**DOI:** 10.3390/ijerph191711146

**Published:** 2022-09-05

**Authors:** Charity Ngoatle, Tebogo M. Mothiba, Modikana A. Ngoepe

**Affiliations:** 1Department of Nursing Sciences, University of Limpopo, Polokwane 0727, South Africa; 2Faculty of Health Sciences, University of Limpopo, Polokwane 0727, South Africa; 3Quality Assurance Unit, University of Limpopo, Polokwane 0727, South Africa

**Keywords:** COVID-19, COVID-19 pandemic, quality, teaching and learning, higher education programmes, document analysis

## Abstract

***Background:*** The COVID-19 pandemic worldwide, has caused a swift change in the higher education system giving way to a rise in instituting multimodal teaching and learning approaches. These approaches have demonstrated an inadequate capacity for multimodal teaching, particularly through online instruction by many institutions. The Department of Higher Education in South Africa did its utmost best to equip the institutions with the required resources to continue with the provision of education. ***Methods:*** A descriptive qualitative research design was followed in the study. The study data source included the South African government’s COVID-19 regulations relating to higher education and training. The purposive sampling method was used to select (8) several government documents relating to the regulation of COVID-19 in higher education and training institutions Document analysis technique was used to collect data from the COVID-19 pandemic regulation documents. ***Results:*** the study showed that most HEIs in South Africa adhere to safety measures, ensure business continuity in teaching and learning, continued with the distribution of National Student Financial Aid Scheme (NSFAS) allowances as allocated by the government, and followed procedures for returning certain categories of students on campuses during the COVID-19 pandemic. ***Conclusions:*** The study has evaluated the support for quality higher education teaching and learning during the pandemic period in South Africa. The study, therefore, recommends the continuing of online teaching as part of blended learning so that institutions could always be ready should situations like this occurs again in the future and above be in sync with digital transformation.

## 1. Introduction

Teaching and learning, community engagement, and research are the three core functions of higher education. Teaching and learning have been proven by the COVID-19 pandemic to be critical in terms of business continuity [1]. The COVID-19 pandemic worldwide has caused a swift change in the teaching and learning approaches giving way to a rise in using multimodal remote learning [1]. The employment of multimodal approaches showed an inadequate capacity resulting in a lack of preparation in online instruction by many institutions [1]. The disruption brought about by the COVID-19 pandemic in the education system affected almost 1.6 billion students in over 200 countries [2]. Most higher education institutions returned to their respective campuses and selected students depending on the nature of their programmes in alert levels while other students continued with online teaching, learning, and assessment [2]. Higher education institutions were forced to come up with and implement methods for maintaining high standards of teaching and learning, including using staff members who could work from home to deliver lectures and find other ways to help students [3]. The Indian government has made online education mandatory at today’s institutions of higher learning [4].

At the beginning of the COVID-19 pandemic in South Africa, the higher education sector became mainly vulnerable and messy [3]. The COVID-19 pandemic also uncovered the inequalities among higher education institutions in South Africa, as some institutions were not ready to move to online teaching and learning due to a lack of adequate technology devices and ICT capacity systems [3]. The COVID-19 pandemic also affected the higher education system in South Africa such that even other associated activities such as graduation ceremonies had to be done virtually [3]. The higher education institutions in South Africa informed by the motto of saving lives and academic years have employed online teaching and learning to a degree that has never been witnessed before [5]. The study sought to evaluate the quality of higher education teaching and learning programmes through the pandemic period in South Africa.

## 2. Methodology

### 2.1. Research Design

A descriptive qualitative research design followed to describe the information provided in the government gazettes related to the phenomenon under study (Polit & Beck, 2022).

### 2.2. Study Site

The context of the study is in the Republic of South Africa’s Higher education and training institutions.

### 2.3. Data source and Sampling

The study data source included all the South African government COVID-19 regulations relating to higher education and training. The purposive sampling method was used to select (8) several government documents relating to the regulation of COVID-19 in higher education and training institutions [6].

#### Inclusion and Exclusion Criteria

The study included all the South African COVID-19 regulation documents, which affect higher education and training. All documents that were not government gazetted were excluded.

### 2.4. Data Collection Procedures

The document analysis technique was used to collect data from the COVID-19 regulation documents and protocols affecting higher education.

### 2.5. Data Analysis

Data were analysed using thematic analysis [7]. Acquainting oneself with the data, generating initial codes, searching for themes, reviewing themes, defining, and naming themes, and producing the report are the six phases of the analytic process followed [7]. Four (4) themes and sixteen (16) sub-themes emerged from the study.

### 2.6. Trustworthiness

Trustworthiness is ensured through authenticity, credibility, dependability, confirmability, and transferability [7].

### 2.7. Ethical Considerations

The study did not require ethical clearance.

## 3. Results

The findings of the study are presented below based on the information contained in the Republic of South Africa government gazette, relating to the provision of education and training in higher education institutions.

Eight (8) documents related to the provision of higher education and training during the COVID-19 pandemic were reviewed. The documents were used to give a directive in terms of providing higher education and training during the pandemic period. Four (4) themes and sixteen (16) sub-themes emerged from the study during data analysis as shown in Table 1 below.

### 3.1. Safety Control Measures

The Republic of South Africa government gazette documents pertaining to the provision of higher education and training have shown that HEIs should adhere to safety control measures during the pandemic while continuing with their core mandate.

#### 3.1.1. Observation of COVID-19 Pandemic Safety Protocol

Government Gazette No. 43414 dated 08 June 2020 requires that HEIs adhere to the following prescriptions on the return of students and staff to campuses:

*Every staff member and student entering the school is subject to daily screening and, if necessary, COVID-19 testing. Surfaces should be continuously cleaned at regular intervals. All visitors to the campus are required to wash their hands and there are hand sanitizers available for usage. High-risk locations like libraries and laboratories should be regularly inspected. the use of national standards and the creation of an individual phase-in approach for each level of risk-adjusted strategy. ensuring that all health precautions are taken while teaching and learning strategies are being implemented, including maintaining a physical distance. Lastly, the institution-level issuance of permissions allows for control over the return to campus.* This implies that an institution cannot continue with its business without first making sure that all the prescribed measures are in place and failure to do so may necessitate an offense.

Government gazette No. 44895 dated 25 July 2021 also requires that: “*Face masks should be compulsory for every individual to wear when in a public setting, and an employer may refuse to permit an employee to enter work premises if the employee is not wearing a face mask. This section also applies to students accessing the academic institution grounds and traveling to the clinical area for experiential learning*”. This implies that individuals were restricted to access any institution of higher education if they had no face mask on.

#### 3.1.2. Isolation and Quarantine Facilities

Government Gazette No. 43414 dated 08 June 2020 requires that: “*all HEIs must establish isolation and quarantine sites, design, and publish policies for any staff or students who exhibit symptoms or test positively*.” This denotes that HEIs were to support students and staff members who needed isolation or quarantine services for the purposes of containing the virus.

#### 3.1.3. Provision of Personal Protective Equipment

Government Gazette No. 43414 dated 08 June 2020 requires that HEIs should: “*Provide masks and other suitable personal protective equipment*”. This signifies that it remained the HEIs’ responsibility to provide protection for its staff and students.

#### 3.1.4. Campuses Access Control

Government Gazette No. 43414 dated 08 June 2020 requires that: “*HEIs offer permits to staff and students so they can enter or exit campuses. To notify students and employees of the phase-in process, each institution should directly engage with them. Measures are in place to prevent the issuance of false permits*”. This means that the HEIs had the duty to control access to, and to notify those staff and students who meet the requirements to gain entry to the HEIs.

### 3.2. Ensuring Continuity of Teaching and Learning through the Pandemic

According to Government gazette No. 43414 dated 08 June 2020, the Republic of South African Higher Education Institutions (HEIs) declared the implementation of a risk-adjusted strategy to phase in the reopening of campus activities. The following aspects are taken into consideration:

#### 3.2.1. International Students

The Government Gazette No. 43414 dated 08 June 2020 indicates that: “*A small number of students, including international students, remained in public institution residences, both on and off campus*”. Furthermore, students were being phased in during alert levels 4 to 2 the Gazette states: “*The international students who had left for their home nations during the lockdown were only allowed to return to institutions when Level 1 of the strategy was disclosed. When these international students returned to university, they were given individualized catch-up plans and supported through distance learning*”. The Gazette thus made provision for the international students who could not go to their respective countries to remain on/off campus residences, whilst those that left were prohibited to return until it was safe to do so.

#### 3.2.2. Vulnerable Students

Based on the Government Gazette No. 44342 dated 29 March 2021 requires that: “*Each institution identifies students who reside in unsuitable study environments or areas with poor connectivity, and wherever necessary, supports their accommodation in residences, whether on or off-campus. These students were assisted in their participation in the institution’s blended learning programs and were given permission to visit campus when needed for practical and other teaching and learning requirements. The gazette further indicates that “institutions must have clear protocols and alternative work and study arrangements for those staff members and students who may be more vulnerable to the virus because of age and/or comorbidities and are, therefore, unable to attend activities in person*”. This entails that, the Department of Higher Education made provision for all students to continue learning taking into account all the factors that may hinder teaching and learning to take place.

#### 3.2.3. Health Sciences Students

The Government gazette No. 43414 dated 08 June 2020 indicates that: *The MBChB students are in their final year of study, followed by final year students enrolled for the Bachelor of Dental Sciences, Bachelor of Dental Surgery, Diploma in Nursing, Bachelor of Science in Nursing, and Bachelor of Veterinary Science as the final year students in programs requiring clinical training return to campuses under alert level 4*. This denotes that only students whose nature of their training programmes required on-site clinical training were prioritized to return to campuses.

#### 3.2.4. Remote Teaching and Learning

The Government Gazette No. 43414 dated 08 June 2020 states: “*Students and employees who have the option of working remotely are urged to do so and should only travel to campus when absolutely necessary, such as to use the library or to complete practical or laboratory work*”. Thus, distance learning was significantly encouraged during the pandemic period in trying to control the spread of the virus.

#### 3.2.5. Unchanged Tuition and Accommodation Contracts

According to Government Gazette No. 43772 dated 05 October 2020, tuition and accommodation fees remained unchanged despite the lengthened academic period. This is evident in Section 5 Sub-Section 5.1 indicating, “*In terms of tuition fees, the 2020 academic year is envisioned as a package, regardless of length*”. According to Sub-Section 5.2, “*the above statement implies that the cost of tuition must be kept the same for the 2020 academic year irrespective of the frame for a student to finish and the delivery method for completion*”. The students, therefore, had an opportunity to study for a prolonged period without being charged extra fees for tuition and accommodation in South Africa.

### 3.3. Continuing NSFAS Allowances

The distribution of NSFAS allowances as allocated by the government continued as normal during the pandemic lockdown period as follows:

#### 3.3.1. Tuition Allowance

According to Government Gazette No. 43772 dated 05 October 2020, Section 5, Sub-Section 5.3, “*NSFAS payments for tuition fees to institutions will be made on the basis of the originally agreed tuition fee*”. The NSFAS continued to pay students their tuition fees as agreed despite the changes brought by the pandemic which decreased the academic financial implications of the pandemic on the students and their families.

#### 3.3.2. Living Allowance

According to Government Gazette No. 43772 dated 05 October 2020, Section 5, Sub-Section 5.9, “*NSFAS allowances have continued to be paid to students during the lockdown period*”. The living allowance was part of the allowances that the students received during the lockdown period, and it helped most students to feed their families as most lost their family income.

#### 3.3.3. Accommodation Allowance

According to Government Gazette No. 43772 dated 05 October 2020, Section 5, Sub-Section 5.10, “*NSFAS will continue to disburse the accommodation allowances to beneficiaries up to the agreed costs for the ten months of the academic year*”. Therefore, students were still expected to pay for their accommodations as agreed with their property owners because the owners also needed income since they were hosting the students’ belongings.

### 3.4. Returning Students to HEIs Campuses

It was observed that returning students back to the South African higher education and training institutions was done based on Regulation No. 652, dated 08 June 2020.

#### 3.4.1. Adjusted Alert Level 4

During the adjusted alert level 4, only a few students meeting the criteria set under level 4 were allowed to return to campuses as follows:

According to regulation No. 652, “*Only a small number of carefully regulated student returns to campuses were permitted. The students included those enrolled in programs including the Bachelor of Dental Science, Bachelor of Dental Surgery, Bachelor of Medicine and Surgery, Diploma in Nursing, Bachelor of Science in Nursing, and Bachelor of Veterinary Science that need clinical training in their final year*”.

#### 3.4.2. Adjusted Alert Level 3

The documents further indicated the students who can be repatriated as follows: According to regulation No. 652 “*On the basis of the established criteria, a maximum of 33% of the total student data source were permitted to return to the schools. The criteria included every student who has already returned during adjusted alert level 4, every student who is in their final year and on track to graduate in May 2021, every student whose programs require clinical training despite the years of study they are in, and every postgraduate student who needs laboratory equipment and other technical equipment to carry out their studies*”. A list of all the students falling under this category has also been provided in the regulation. “*Students should be phased in only if the clinical training platforms are prepared and have sufficient space to accommodate the students while adhering to safety protocols*”.

The regulations further stipulated, “*Until such time as it is deemed safe to return to campus, all other students are supported* via *remote multimodal teaching, learning, and assessment.” Students in private higher education institutions with fewer than 50 students may be exempted from returning to their campus pending Department approval, along with a plan for the return of staff and students that complies with the approved COVID-19 guidelines*”. This signifies that a thorough safety assessment was to be made by the HEIs to phase in students who met the criteria to return to campus under this level.

#### 3.4.3. Adjusted Alert Level 2

The adjusted alert level 2 covers the following criteria for the return of students to campus: “*a maximum of 66% of the total student data source, including students who have already been phased-in during levels 4 and 3, students who were unable to be accommodated due to campus carrying capacity, students who need to complete the academic year through practical placements, experiential learning, or workplace-based learning, and first-year students in all undergraduate programs*”. This level thus required HEIs to continue with their thorough assessment to identify students who met the criteria to be phased under alert level 2.

#### 3.4.4. Adjusted Alert Level 1

Phasing-in of students under adjusted alert level one was done based on the following criteria: “*All student population is permitted to return to campus; however, physical separation and health protocols continue to be in effect, social solidarity is still required, and only those international students who left South Africa during the lockdown period with permission for international travel may return to campus*”. This implies that all registered students in the HEIs could return to campus, however, COVID-19 protocols were still expected to be adhered to.

## 4. Discussion

The study revealed that there were safety control measures in the place prescribed by the Department of Higher Education and Training to ensure continuity of quality teaching, learning, and assessment in higher education institutions (HEIs). The safety control measures included observation of COVID-19 pandemic safety protocols, establishing isolation and quarantine facilities, provision of personal protective equipment, and establishing campus access control. The findings supported by the study conducted in two rural universities in South Africa indicate that there were measures in place in response to the COVID-19 pandemic regulations and protocols approved by the world health organization [8]. The results meant that some South African Department of Higher Education and Training like other countries, took efforts to establish COVID-19 pandemic safety measures so that HEIs could continue providing education to students during the new normal.

The study also showed that the South African Department of Higher Education and Training set up regulations and control measures for HEIs to continue with teaching and learning during the pandemic. This encompassed the following: remote teaching for international students, making provision for vulnerable students to access learning, supporting health sciences students to continue with their experiential learning, remote teaching and learning for the other students, and unchanged tuition and accommodation contracts for the NSFAS deserving students. Similar findings have been observed where students and lecturers used online teaching and learning for continuity of education [9]. Another study also supported the finding that a high number of South African students indicated an improvement in digital literacy due to online teaching and learning [10]. This was because online teaching and learning was the best available method of teaching during the pandemic. South African students further indicated that they had a proper digital learning environment at home [10]. This might be a result of the majority of families accepting remote working and studying as the new norm brought about by COVID-19.

Moodley [11] added that despite not studying consistently prior to the pandemic, students improved their learning strategies and advanced their knowledge and skills during the pandemic. His study showed that learning during the pandemic positively affected students’ independent learning strategies, but it was unable to determine whether the change in teaching methodology had any impact on better student performance [11].

During the COVID-19 pandemic in South Africa, students in institutions of higher learning continued getting their NSFAS allowances. The allowances included tuition allowance, living allowance, and accommodation allowances. The findings are supported by the study indicating that the Minister for Higher Education and Training confirmed that all the students who registered for the 2020 NSFAS application, will get digital devices and data bundles [12]. The minister further indicated that the allowances covered under NSFAS that is, tuition, laptops, books, living allowances, pocket money, and accommodation allowances will continue issued as planned [12]. These measures were put in place so that provision of higher education should not be hampered by the COVID-19 pandemic. Another reason for the NSFAS allowances was to make sure that students receive resources to facilitate remote learning. Before the COVID-19 pandemic, students in higher education institutions did not receive data to access teaching and learning. However, due to the instituting of blended learning post the pandemic, the Minister of Higher education indicated that the amount of data required for online access for teaching and learning, as well as assessments, remained high across the system. He further said that, for all university graduate (UG) students, the overall system average is 86%. As a result, the department will collaborate with the Minister of Communication and Digital Technologies to establish a dedicated continuum for higher education and training, to enable effective access, sustainment, and use of digital technologies in support of education and training [13].

Lastly, the results outlined the phasing-in of students to HEIs’ campuses under adjusted alert level 4, adjusted alert level 3, adjusted alert level 2, and adjusted alert level 1. These results are affirmed by results indicating that universities planned for a specific number of students to return to campus within the prescribed COVID-19 safety regulations [12]. As much as most students were continuing with remote teaching and learning, both the lecturers and students experienced some challenges which were not evident before the COVID-19 pandemic. Students struggled with applying advanced settings to some software and programs that were not prominent during face-to-face instruction [14]. Alex [14] therefore indicated that it is necessary to reframe, train and develop both students and lecturers in online teaching and learning.

## 5. Conclusions

The study has evaluated the provision of quality higher education teaching and learning programmes through the pandemic period in South Africa. It is evident from the study results that despite the challenges brought by the COVID-19 pandemic to South African higher education institutions, the Department of higher education and training provided supportive measures for HEIs to continue proving quality teaching and learning. The results, however, are cognisant of the hiccups brought by transitioning to remote teaching and learning. The study, therefore, recommends the continuing of online teaching as part of blended learning so that institutions could always be ready should situations such as this occurs again in the future over above be responsive and relevant to digital transformation.

## Figures and Tables

**Table 1 ijerph-19-11146-t001:** Themes and sub-themes of the study results.

Themes	Sub-Themes
Safety control measures	1.1 Observation of COVID-19 pandemic safety protocols 1.2 Isolation and quarantine facilities 1.3 Provision of personal protective equipment 1.4 Campuses access control
2. Ensuring continuity of teaching and learning through the pandemic	2.1 International students 2.2 Vulnerable students 2.3 Health sciences students 2.4 Remote teaching and learning 2.5 Unchanged tuition and accommodation contracts
3. Continuing NSFAS allowances	3.1 Tuition allowance 3.2 Living allowance 3.3 Accommodation allowance
4. Phasing-in of students to HEIs campuses	4.1 Adjusted alert level 4 4.2 Adjusted alert level 3 4.3 Adjusted alert level 2 4.4 Adjusted alert level 1

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
