# Peer review of "Navigating through COVID-19 Pandemic Period in Implementing Quality Teaching and Learning for Higher Education Programmes: A Document Analysis Study"

_ijerph, 2022, doi:10.3390/ijerph191711146_

Round 1

Reviewer 1 Report

This work covers a relevant and interesting topic but there is major work to be done before it can be considered for publication. I've outlined my specific concerns:

Abstract: The first sentence of the abstract is wordy and repetitive. It could be shortened and clarified. Guidelines can’t be considered a ‘population’ but it can be viewed as a data source. You can’t infer that higher education institutions will adhere to the guidelines and follow the recommendations, just because those documents exist.

Introduction: You shouldn’t mention the three core functions of HE without naming them.

Methodology: Documents can’t be considered a population, but they can be considered a data source. The terminology just needs to be corrected. There needs to be a bit more elaboration around each of the points, particularly in relation to how the thematic analysis was done.

Results: There needs to be more discussion and contextualisation around the results, rather than simply repeating what the documents say across a range of topics.

Discussion: It sounds as if two South African universities were investigated to determine their compliance to the regulations. If that is the case, this part needs to be included in the methodology and a description of how it was done. Without this, some of the implications don’t follow. Just because there is a regulatory framework, it’s impossible to say whether South African higher education institutions are following that. For example, page 8, line 250 says: ‘The study also showed that South African HEIs ensured continuity of teaching and learning through the pandemic.’ This can’t be ascertained through the work that was done.

Author Response

Thank you for the valuable comments. Changes have been effected as follows:

1. Comment 1: Abstract: The first sentence of the abstract is wordy and repetitive. It could be shortened and clarified. Response: effected on lines 15 - 17. Guidelines can’t be considered a ‘population’ but they can be viewed as a data source. Response: effected on line 21. You can’t infer that higher education institutions will adhere to the guidelines and follow the recommendations, just because those documents exist. Response: effected on line 29. 

2. Comment 2: Introduction: You shouldn’t mention the three core functions of HE without naming them. Response 2: effected on line 38.

3. Comment 3: Methodology: Documents can’t be considered a population, but they can be considered a data source. Response: effected on line 72. The terminology just needs to be corrected. There needs to be a bit more elaboration around each of the points, particularly in relation to how the thematic analysis was done. Response: effected on lines 85 - 87.

4. Comment 4: Results: There needs to be more discussion and contextualisation around the results, rather than simply repeating what the documents say across a range of topics. Response 4: effected on lines 111-260.

5. Comment 5: Discussion: It sounds as if two South African universities were investigated to determine their compliance to the regulations. If that is the case, this part needs to be included in the methodology and a description of how it was done. Without this, some of the implications don’t follow. Just because there is a regulatory framework, it’s impossible to say whether South African higher education institutions are following that. For example, page 8, line 250 says: ‘The study also showed that South African HEIs ensured continuity of teaching and learning through the pandemic.’ This can’t be ascertained through the work that was done. Response 5: effected on lines 111-303.

Reviewer 2 Report

The article does not explore the consequences of the documents it has dealt with and does not produce specific examples. An accurate analysis (with data and examples) is to be expected.

The paragraph 2 and 3 need more detail

Author Response

1. Comment: The article does not explore the consequences of the documents it has dealt with and does not produce specific examples. An accurate analysis (with data and examples) is to be expected. Response: effected on line 111-303.

2. Comment: The paragraph 2 and 3 need more detail. Response: effected on line 15 to 51.

Round 2

Reviewer 1 Report

This paper is much improved with the changes effected by the author/s.

I'm not sure why there is reference to Indian government mandates around online learning on page 2, line 51.

It is not clear to me how the author/s know that South African HEIs complied with the requirements. This needs to be covered off in the methodology. I think the author/s can say whether or not the HEI in which they worked is compliant. But how do they know about the others? Did they have colleagues? News reports? Bulletins? Journal articles? The explanation needs to be explicit. Page 7, line 272 and line 276.

Author Response

Thank you for the valued comment. Please find the response to the comment.

comment: It is not clear to me how the author/s know that South African HEIs complied with the requirements. This needs to be covered off in the methodology. I think the author/s can say whether or not the HEI in which they worked is compliant. But how do they know about the others? Did they have colleagues? News reports? Bulletins? Journal articles? The explanation needs to be explicit. Page 7, line 272 and line 276.

response: sentences were restructured to improve the meaning. see page Page 7, line 271 to line 281.

Reviewer 2 Report

The second version of the paper is better than the first one. In any case it's not clear the link between the official documents and the quality teaching and learning and the consequences post-Covid period.

We suggest to increase the research with some example of actually activities and actions promoted after the Covid period

Author Response

Thank you for the comment.

Comment: The second version of the paper is better than the first one. In any case it's not clear the link between the official documents and the quality teaching and learning and the consequences post-Covid period.

We suggest to increase the research with some example of actually activities and actions promoted after the Covid period.

response: the comment was effected on page 7, line 289 - 293. then on page 8, line 307 - 313, lastly, on page 8, line 318 - 312 respectfully.